# Determinants of subjective total athletic ability

**Sho Ito[ID][1]\*, Keishi Soga[2], Kouki Kato[1]**

**1** Physical Education Center, Nanzan University, Nagoya, Aichi, Japan, **2** Smart Aging International Research Center, Tohoku University, Sendai, Miyagi, Japan

\* shoito@nanzan-u.ac.jp

## Abstract

The term "good motor skill" is often discussed in everyday contexts and when observing sports; however, its definition remains elusive, and the associated factors are not well understood. Therefore, in this cross-sectional study, we investigated the determinants of subjective total athletic ability, defined as the sum of subjective athletic abilities across 11 sports disciplines. A sample of 406 undergraduate students completed a questionnaire to evaluate their perceived athletic prowess in various sports, as well as assessments of their personality traits, family background, and sports performance. The analysis revealed correlations between the perceived general athletic ability and specific abilities in soccer, volleyball, basketball, and short-distance racing. Furthermore, linear model analyses indicated a positive association between perceived total athletic ability and personal characteristics such as grit, resilience, and a growth mindset. Factors such as recreational activities in elementary school, sibling structure, prior athletic experience, parental athletic ability, family income, external evaluations of motor skills, and age at first walking also appeared to influence perceived total athletic ability. These results imply that a blend of internal and external factors may shape subjective athletic ability. However, future studies should investigate the causal connections among these factors to deepen our understanding of the concept and its influencers.

## 1. Introduction

The awareness of subjective athletic ability is crucial for an individual's participation in physical activities and sports competitions. A strong perception of athletic competence fosters engagement, whereas feeling inferior in skills may discourage participation. Participation in sports and physical activities offers numerous health benefits [1] and can considerably influence students's academic records [2]. Despite frequent references to "He/She is athletic" in daily conversations and sports viewership, the definition is remains vague. This term is commonly understood as an overall evaluation of physical strength and athletic capability as well as the ability to excel across various sports. Praise for athletic ability generally indicates a superior sports

**Data availability statement:** All relevant data are within the manuscript and its Supporting information files.

**Funding:** We would like to express our sincere gratitude to the Nanzan University Pache Research Subsidy I-A-2 for the 2024 academic year and JSPS KAKENHI Grant Number JP23K10623 for funding this research. 1 Initials of the authors who received each award; Kouki Kato Grant numbers awarded to each author ; JP23K10623 The full name of each funder: Japan Society for the Promotion of Science URL of each funder website; https://www.jsps.go.jp/j-grantsinaid/ 2 Initials of the authors who received each award; Kouki Kato Grant numbers awarded to each author ; Nanzan University Pache Research Subsidy I-A-2 for the 2024 academic year The full name of each funder: Nanzan University URL of each funder website; https://www.nanzan-u.ac.jp/.

**Competing interests:** The authors have declared that no competing interests exist.

performance. However, the elements that contribute to motor skill proficiency are not clearly defined. Therefore, elucidating the fundamental aspects that influence motor-skill development will help establish more precise criteria for this term.

In physical education and sports psychology, research on "motor competence" has been widely conducted on children and adolescents regarding athletic ability and its evaluation [3–7]. Motor competence is a measure of self-perception of competence or confidence in physical activity [8]. Confidence resulting from successful exercise experiences (a sense of motor competence) plays a crucial role in improving athletic ability and personality development. This can be understood through the framework of self-efficacy theory proposed by Bandura [9], which identifies prior successes, social support, and observational learning as key predictors for the development of self-efficacy. Thus, motor competence can be seen as a specific manifestation of self-efficacy that promotes further achievement [10].

Studies have also been conducted on the relationship between lifestyle, family environment, athletic ability, and exercise habits [11–16]. In this study, athletic ability is defined as the general concept of physical capability, which includes components such as strength, speed, coordination, and endurance. These studies suggest that motor competence and athletic ability are influenced by various lifestyle factors. While numerous studies have focused on children and adolescents, research on college students remains limited. In addition, most previous studies focused on physical fitness tests and specific movements [17–19], with no studies having explored abilities in various sports disciplines.

Relative age can influence physical fitness factors and characteristics in different sports [20]. These effects suggest that differences in physical and mental development among students in the same grade affect academic achievement [21] and athletic performance [20]. The relative age effect is prominent in major sports such as baseball and soccer, whereas no effects were observed for relatively minor sports such as handball and badminton [22]. Additionally, the nature of the sport, whether it is a ball game, team sport, or individual sport, may also affect subjective athletic ability. Therefore, differences in sports experience may affect a sense of motor competence. Furthermore, based on studies conducted among university students, it was found that women with higher perceptions of motor competence also exhibited greater physical fitness. However, no significant difference in physical fitness was observed between men with high and low perceptions of motor competence. The authors suggest that this may be due to the influence of sports club experience and specialized athletic skills, which are not assessed by general physical fitness tests. This test focuses on basic motor abilities such as jumping, running, and throwing [4]. This indicates that differences in sports experience may significantly influence subjective athletic ability, or that individuals can estimate their physical abilities with relative accuracy.

Successful sports experiences require perseverance, as athletes often encounter failures and setbacks along the way. Personality traits are vital for good athletic performance in sports [23]. The personality trait, grit, which is the ability to persevere, is often used as an indicator of the capacity to achieve long-term goals despite

difficult situations [24]. Athletes with higher grit ability tend to spend more time actively responding to competition, leading to improvements in sports skills [25]. Additionally, resilience and a growth mindset are crucial in overcoming challenges. Resilience is the ability to bounce back from difficult situations [26], and a growth mindset is the belief that talent can be developed consistent efforts to improve one's ability [27]. Higher levels of psychological factors such as grit, resilience, and a growth mindset may be associated with greater persistence in sports, which in turn is likely to enhance athletic performance and develop athletic ability and motor competence.

Therefore, this study aimed to explore the relationship between subjective total athletic ability, defined as the sum of subjective athletic abilities across 11 sports disciplines, and selected factors such as personality traits, sibling composition, and athletic performance among college students. Given the exploratory nature of this study, we examined potential influencing factors on subjective athletic ability from various perspectives, acknowledging the limitations inherent in self-reported measures and the challenges of defining subjective athletic ability with precision. We hypothesized that subjective athletic ability among college students is influenced by personality traits, family environment (e.g., sibling composition), and prior athletic performance. For example, students with a history of participation in competitive sports may report higher subjective athletic ability compared to those without such experience.

## 2. Materials and methods

### 2.1. Participants

Data were collected from 421 survey participants who were students at Nanzan University and their acquaintances. The study's outline and purpose were fully explained to the participants, and consent was obtained from 409 participants. To consider the influence of sex in this study, 406 of those who gave consent and whose sex was specified were included in the analysis (mean age: 19.31 years, SD: 0.95; female: 221, male: 185). This study was approved by the Ethics Committee for Human Subjects of Nanzan University (Approval No. 23–072). In this study, informed consent was obtained from participants via Google Forms (Alphabet Inc., U.S.A.). For minors involved in this study, informed consent was obtained directly from the participants, while the requirement for parental or guardian consent was waived by the ethics committee. The data for this study were collected between December 5, 2023, and December 18, 2023.

### 2.2. Questionnaire

#### 1) Face sheet

The respondents were asked to choose from a list of options for age, birth month, and sex. Regarding sibling composition, participants were asked to select all that apply from the following groups: "I have an older brother within 2 years," "I have an older brother of ≥ 3 years," "I have an older sister within 2 years," "I have an older sister of ≥ 3 years," "I have a younger brother within 2 years," "I have a younger brother of ≥ 3 years," "I have a younger sister within 2 years," "I have a younger sister of ≥ 3 years," and "I am an only child." The participants were asked to select their father's and mother's annual income from the following options: "don't know," "< 3 million yen," "3.01 million – 4 million yen," "4.01 million – 5 million yen," "5.01 million – 6 million yen," "6.01 million – 7 million yen," "7.01 million – 8 million yen," "8.01 million – 9 million yen," "9.01 million – 10 million yen," and " ≥10.01 million yen ", respectively. The participants were asked to select the most typical ways of spending their leisure time during the early elementary school years from the following categories: "games," "studying," "reading," "music (singing and playing musical instruments)," and "exercise (physical play)." Participants were asked to answer the same questions on how they spent their leisure time in the middle and upper elementary school grades.

#### 2) Sports experience

Initially, the participants were asked to choose "Yes" or "No" to the question, "Have you ever learned any physical exercise or sports? " Those who responded "yes" were asked additional questions regarding the number of events they had

experienced, the age at which they started playing sports, specialized events, and athletic performance. The participants were asked to select the number of sports events they had experienced from the options "1," "2," "3," "4," and "≥5 ". Participants were also asked to select the age at which they started learning sports from the following age groups: < 3, 3, 4, 5, 6, 7, 8, 9, 10, 11, 12, 13, 14, 15, and ≥16 years. Additionally, they were asked to select the sport in which they had the longest experience from the following categories: baseball, soccer, rugby, basketball, tennis, badminton, volleyball, track and field (short distance), track and field (long distance), swimming, table tennis, kendo, judo, and gymnastics. Regarding the competition results, the participants were asked to select from the following levels: "Prefectural Qualifier Level," "Prefectural Tournament Participant Level," "Prefectural Tournament Placement Level," and "National Tournament Participant Level. "

### 3) Subjective athletic ability

Subjective general athletic ability, which refers to an individual's self-assessment of their overall physical competence across various types of physical activities and sports, subjective athletic ability in each sport (volleyball, basketball, badminton, tennis, baseball, table tennis, soccer, gymnastics, short-distance race, and long-distance race), father's athletic ability, and mother's athletic ability were assessed using an 11-point scale (1 = lowest athletic ability and 11 = highest athletic ability). The participants were asked to select the age at which they became aware of their athletic ability (superiority or inferiority) from the following age groups: 4, 5, 6, 7, 8, 9, 10, 11, 12, 13, 14, 15, 16, 17, 18, and ≥ 19 years. Regarding evaluation by others, the participants were asked, "Have you ever been told by others that you are a good athlete? They were to choose from "often," "sometimes," "not often," and "never." The participants were asked to select their answer from "very much faster than the average person," "a little faster than the average person," "about average," "a little slower than the average person," or "very much slower than the average person" to the question "What is the average speed at which you learn a movement (technique) for the first time?"

### 4) Personality characteristics

For the evaluation of grit, we used the Japanese version of the Short Grit scale [28], which comprises eight items. The original version of this scale was developed in English [29], and Nishikawa et al. assessed the validity of the questionnaire translated into Japanese, reporting that it retained a reliability, and basic validity as the original short Grit scale [28]. Responses were recorded on a five-point scale, with one indicating "not applicable" and five indicating "applicable." In this study, the total score of each item was used as the grit value for data analysis. The Cronbach's alpha coefficient for this scale was 0.7.

For the evaluation of resilience, we used the Japanese version of the Brief Resilience Scale [30], which was originally developed in English [26], and Tokuyoshi and Moriya found that the Japanese version of the scale demonstrated acceptable construct validity, supporting its use for assessing relevant constructs in Japanese populations [30]. The scale includes six items, and participants were instructed to respond using a five-point scale (1 = not applicable at all, 2 = not very applicable, 3 = undecided, 4 = somewhat applicable, and 5 = quite applicable). In this study, the total score for each question item was used as the analytical data for the resilience value. Cronbach's alpha coefficient for this scale was 0.88.

We used the Intelligence Scale as an indicator to measure growth mindset. Ten items from the Intelligence Scale were used in this study [31]. Examples of the items include "Intelligence is an innate ability determined at birth" and "I believe that I can become smarter through effort." Responses were recorded on a five-point scale, with one indicating "not true" and five indicating "very true." In this study, the total score of each question item was used as analytical data for the intelligence view. Cronbach's alpha coefficient for this scale was 0.77.

## 2.3. Data analysis

We calculated the required sample size using G*Power (version. 3.1) with the following parameters: a relatively small effect size ($f^2 = 0.04$), α = 0.05, power = 0.80, and 8 predictors (the maximum number of variables in our model). The

analysis indicated that a total sample size of 384 was required. Our actual sample size (N = 406) exceeded this requirement, confirming that it was sufficient to detect the hypothesized effects.

Data analyses were conducted using Python (version. 3.11.1), and R (version. 4.3.2). As part of the data preprocessing, responses that did not answer the question, such as "I don't know," were treated as missing values. In this study, a listwise exclusion method was used without completing missing values. The data used in each regression model analysis are listed respectively (Tables 1 and 2). The collected responses were converted to numbers, and dummy variables were used in the analysis when categorical data were included in the regression model. All models showed an association with subjective total athletic ability based on a nonstandardized coefficient (B).

A correlation analysis was conducted using Pearson's product-rate correlation coefficient to examine the relationship between an individual's subjective general athletic performance and their subjective athletic abilities in each sport.

To identify the factors influencing gross motor skills, we conducted a linear regression analysis using an individual's subjective total motor skill, which is the total subjective athletic abilities score for each discipline, as the objective variable. The lm () function of R was used to perform linear regression analysis. Nine linear regression analysis models were created, including different explanatory variables, with the subjective total exercise capacity as the objective variable. The covariates in all models were sex, age, and birth month. Japan uses a unique annual age-grouping policy for sports and education, effective from April 1 to March 31 of the following year. Consequently, for the birth month, the numbers were adjusted such that April births began at 1 and March births ended at 12. Model 1 included three personality traits: grit, resilience, and intelligence. Grit: For reliability, Cronbach's α was.70 and McDonald's ω total was.76. Resilience: For reliability, Cronbach's α was.88 and McDonald's ω total was.89. Intelligence Scale: For reliability, Cronbach's α was.77 and McDonald's ω total was.77. Model 2 included leisure-time activities in elementary schools categorized by grade level. A dummy variable was used for leisure time, with exercise as the criterion variable. Model 3 included sibling composition. A dummy variable was used, and the reference category was set as an only child. Model 4 included the presence or absence of exercise experience. Model 5 incorporated variables such as the number of sports practiced over a year, age at first engagement with sports, and competition performance. A dummy variable was used for competition performance and the reference category was the qualification level required for the prefectural tournament. Model 6 included the combined annual household income of the parents (10, 000 yen). Model 8 included the subjective athletic performances of fathers and mothers. Model 9 included the age at which the participants became aware of the superiority of their motor skills, the frequency of being told by others that they are "athletic," and the speed at which they learned the movements (movements/techniques) they were performing for the first time. A dummy variable was used for the frequency of being told words, such as "you have good motor skills," and the reference category was "never been told." The speed of learning a movement (movement/technique) for the first time was also used as a dummy variable, with the reference category being "very slow compared with the average person." The coefficients of determination were listed as $R^2$ values of the adjusted coefficients of determination.

## 3. Results

The correlations between subjective general athletic ability and athletic ability in each sport and their relationships are shown in Fig 1. The analysis revealed that the subjective general athletic ability was strongly correlated with soccer, volleyball, basketball, and short-distance racing. Tables 1 and 2 present the results of the linear model analysis. The relationship between the explanatory variables used in each model and subjective total athletic ability is illustrated in Figs 2 and 3. Model 1 demonstrated that higher scores for grit, resilience and growth mindset were associated with greater subjective total athletic ability ($F$ (6, 399) = 14.73, $R^2$ = 0.17, $p$ < 0.001; Fig 2). Model 2 indicated that leisure activities, such as games and studying in the middle grades, along with games, reading, and music in the higher grades, were negatively correlated with subjective total athletic ability. ($F$ (15, 390) = 10.24, $R^2$ = 0.25, $p$ < 0.001, Fig 3). In Model 3, where sibling composition was an explanatory variable, a positive association was observed between youngest siblings and subjective total athletic

**Table 1. Results of multiple regression analysis in subjective total athletic ability (Models 1–4).**

| | Model 1 | | Model 2 | | Model 3 | | Model4 | |
|---|---|---|---|---|---|---|---|---|
| Variables | B | 95% CI | B | 95% CI | B | 95% CI | B | 95% CI |
| (Intercept) | -12.903 | [-46.215, 20.410] | 44.231** | [13.565, 74.896] | 15.626 | [-18.512, 49.763] | 14.993 | [-17.506, 47.491] |
| Sex | 7.638*** | [4.504, 10.772] | 5.966*** | [2.823, 9.110] | 8.197*** | [4.894, 11.501] | 7.206*** | [4.021, 10.391] |
| Age | 1.544+ | [-0.153, 3.241] | 0.841 | [-0.787, 2.470] | 1.545+ | [-0.259, 3.349] | 1.159 | [-0.578, 2.896] |
| Birth Month | 0.135 | [-0.329, 0.599] | 0.039 | [-0.410, 0.487] | 0.090 | [-0.399, 0.580] | 0.017 | [-0.456, 0.490] |
| **Personality** | | | | | | | | |
| Grit | 0.404** | [0.120, 0.687] | | | | | | |
| Resilience | 0.586*** | [0.340, 0.832] | | | | | | |
| Intelligence | 0.409** | [0.160, 0.658] | | | | | | |
| **Leisure Time** | | | | | | | | |
| Lower Grade | | | | | | | | |
| Game | | | -0.221 | [-5.268, 4.825] | | | | |
| Reading | | | -5.934+ | [-11.984, 0.117] | | | | |
| Music | | | -4.314 | [-14.234, 5.606] | | | | |
| Study | | | 1.840 | [-7.975, 11.655] | | | | |
| Middle Grade | | | | | | | | |
| Game | | | -8.297*** | [-13.194, -3.399] | | | | |
| Reading | | | -5.972+ | [-12.628, 0.684] | | | | |
| Music | | | -3.989 | [-14.290, 6.311] | | | | |
| Study | | | -15.317** | [-24.993, -5.642] | | | | |
| Higher Grade | | | | | | | | |
| Game | | | -7.877*** | [-11.987, -3.767] | | | | |
| Reading | | | -10.659*** | [-16.790, -4.529] | | | | |
| Music | | | -11.770** | [-20.035, -3.505] | | | | |
| Study | | | -4.645 | [-10.616, 1.326] | | | | |
| **Birth order** | | | | | | | | |
| Only Child | | | | | 3.932 | [-3.595, 11.458] | | |
| Oldest Child | | | | | 5.409 | [-2.815, 13.632] | | |
| Middle Child | | | | | 8.229* | [0.689, 15.768] | | |
| Youngest Child | | | | | | | | |
| **Sport History** | | | | | | | 16.109*** | [10.641, 21.577] |
| | Model 1 | | Model 2 | | Model 3 | | Model4 | |
| Num.Obs. | 406 | | 406 | | 406 | | 406 | |
| R² | 0.181 | | 0.283 | | 0.090 | | 0.143 | |
| R² Adj. | 0.169 | | 0.255 | | 0.076 | | 0.134 | |
| Cohen's f² | 0.222 | | 0.394 | | 0.098 | | 0.167 | |

***: $p < 0.001$, **: $p < 0.01$, *: $p < 0.05$, +: $p < 0.1$.

ability ($F_{(6, 399)} = 6.54$, $R^2 = 0.08$, $p < 0.001$; Fig 3). The results of Model 4 demonstrated a positive association between sports experience and subjective total athletic ability ($F_{(4, 401)} = 16.69$, $R^2 = 0.13$, $p < 0.001$, Fig 3). The results of Model 5 showed that the number of sports learned for > 1 year and athletic performance were related to subjective total athletic ability, but not the age at which the first exercise or sport was learned ($F_{(8, 360)} = 11.33$, $R^2 = 0.18$, $p < 0.001$, Figs 2 and 3). Model 6, which examined the association with annual household income, revealed a positive association between subjective total athletic ability and parental household income of the parents ($F_{(4, 145)} = 7.69$, $R^2 = 0.15$, $p < 0.001$;

**Table 2. Results of multiple regression analysis in subjective total athletic ability (Models 5–8).**

| Variables | Model 5 B | Model 5 95% CI | Model 6 B | Model 6 95% CI | Model 7 B | Model 7 95% CI | Model 8 B | Model 8 95% CI |
|---|---|---|---|---|---|---|---|---|
| (Intercept) | 13.937 | [-17.824, 45.698] | -22.322 | [-77.827, 33.184] | -4.408 | [-38.350, 29.533] | 8.773 | [-16.553, 34.099] |
| Sex | 3.579* | [0.366, 6.793] | 9.777*** | [4.140, 15.414] | 9.957*** | [6.717, 13.196] | 6.834*** | [4.380, 9.288] |
| Age | 1.363 | [-0.304, 3.031] | 2.451 | [-0.505, 5.408] | 1.685+ | [-0.080, 3.450] | 0.949 | [-0.367, 2.265] |
| Birth Month | 0.078 | [-0.380, 0.536] | 0.799+ | [-0.001, 1.599] | 0.104 | [-0.381, 0.590] | -0.173 | [-0.533, 0.186] |
| Sport Kind | 3.164*** | [1.733, 4.595] | | | | | | |
| **First Year** | 0.451 | [-0.180, 1.082] | | | | | | |
| **Competition** | | | | | | | | |
| Prefectural Qualifier Level | 8.925*** | [5.187, 12.663] | | | | | | |
| Prefectural Tournament Level | 13.326*** | [8.321, 18.332] | | | | | | |
| Prefectural Award Level | 8.452** | [2.555, 14.349] | | | | | | |
| National Tournament Level | | | | | | | | |
| **Income** | | | 0.018** | [0.007, 0.028] | | | | |
| **Parents' Ability** | | | | | | | | |
| Father Ability | | | | | 1.764*** | [0.962, 2.566] | | |
| Mother Ability | | | | | 1.616*** | [0.920, 2.312] | | |
| **Others** | | | | | | | | |
| Recognition Age | | | | | | | 0.123 | [-0.279, 0.525] |
| Compliment | | | | | | | | |
| Not often said | | | | | | | 9.439*** | [5.316, 13.561] |
| Sometimes said | | | | | | | 17.317*** | [13.361, 21.274] |
| Often said | | | | | | | 25.976*** | [21.338, 30.613] |
| Motor Learning Speed | | | | | | | | |
| Little Slow | | | | | | | 7.138** | [1.784, 12.491] |
| Average | | | | | | | 10.847*** | [5.318, 16.375] |
| Little Fast | | | | | | | 13.376*** | [7.704, 19.048] |
| Too Fast | | | | | | | 15.023*** | [7.935, 22.110] |
| | Model 5 | | Model 6 | | Model 7 | | Model 8 | |
| Num.Obs. | 369 | | 150 | | 375 | | 406 | |
| R2 | 0.201 | | 0.175 | | 0.193 | | 0.519 | |
| R2 Adj. | 0.183 | | 0.152 | | 0.182 | | 0.506 | |
| Cohen's f² | 0.252 | | 0.212 | | 0.239 | | 1.081 | |

***: $p<0.001$, **: $p<0.01$, *: $p<0.05$, +: $p<0.1$.

Tables 1 and 2 present the models used in multiple regression analyses. Sex, age, and birth month were used as covariates.

B: Standardizing Coefficient, CI: Confidence Interval.

Fig 2). Model 8 indicated that the subjective athletic ability of fathers 'and mothers' was associated with increased individual subjective total athletic ability ($F_{(5, 369)} = 17.66$, $R^2 = 0.18$, $p<0.001$, Fig 2). Model 9 revealed that individuals frequently described by others as "superior athletes" exhibited higher subjective total athletic ability ($F_{(11, 394)} = 38.73$, $R^2 = 0.51$, $p<0.001$, Figs 2 and 3). Additionally, using the same model, it was observed that a faster initial learning speed of a movement (technique or skill) was positively associated with greater subjective total athletic ability.

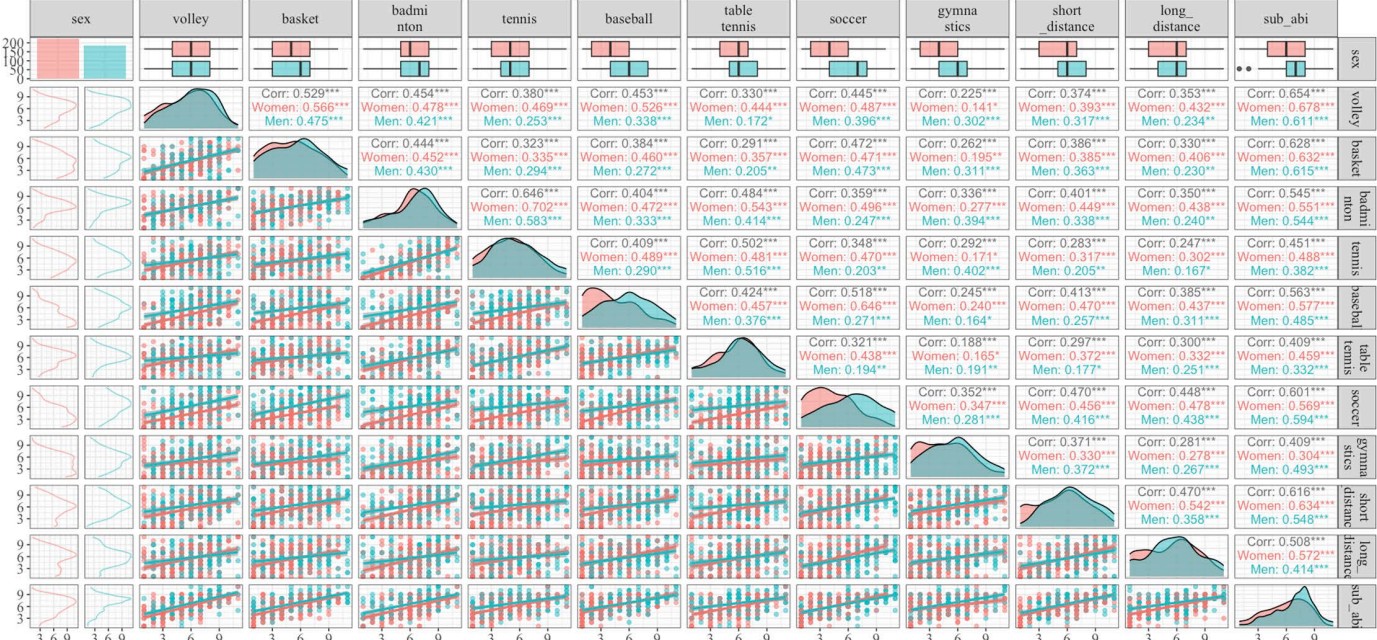

**Fig 1. Correlation between subjective general athletic ability and subjective athletic ability in each sport.** For dots, lines, and box plots, blue indicates males and red indicates females, respectively. The numbers in the upper right-hand corner indicate correlation coefficients. ***: $p < 0.001$, **: $p < 0.01$.

Table 3 presents the diagnostic test results for the assumptions of each model. The Durbin-Watson test was conducted to assess autocorrelation in the residuals, with values ranging from 1.82 to 2.08. Most models showed no autocorrelation, except for Model 7, which indicated marginal autocorrelation. Additionally, the Breusch-Pagan test was performed to evaluate homoscedasticity, and all models demonstrated non-significant results ($p > .05$), confirming this assumption. For Model 7, the Durbin-Watson test was repeated 10 times, with all p-values exceeding.05, supporting the independence assumption.

The horizontal axis shows each category of explanatory variables, and the vertical axis shows total athletic ability. The dots in the figure represent individual observations, and each color represents a different leisure activity.

## 4. Discussion

Awareness of proficient motor skills may encourage individuals to engage in physical activity, which could be crucial to their athletic careers, particularly during their youth. However, the concept of motor skills remains ambiguous and lacks a clear definition. Motor skills are often referred to as an individual's overall athletic ability; however, this term is subjective and may encompass various aspects of physical competence. In this study, we aimed to conduct a questionnaire survey among college students to investigate the relationship between their subjective general athletic ability (commonly referred to as motor skills) and their subjective athletic ability in various sports, personality traits, sibling composition, socio-economic status, and athletic performance. We hypothesized that subjective athletic ability among college students is influenced by personality traits, family environment (e.g., sibling composition), and prior athletic performance. Subjective general athletic ability was highly correlated with subjective athletic ability in short-distance races such as volleyball, basketball, and soccer. Furthermore, various elements such as personality traits, leisure-time activities, and sibling composition were associated with total athletic ability. These findings support our hypothesis that various factors are associated with subjective athletic ability.

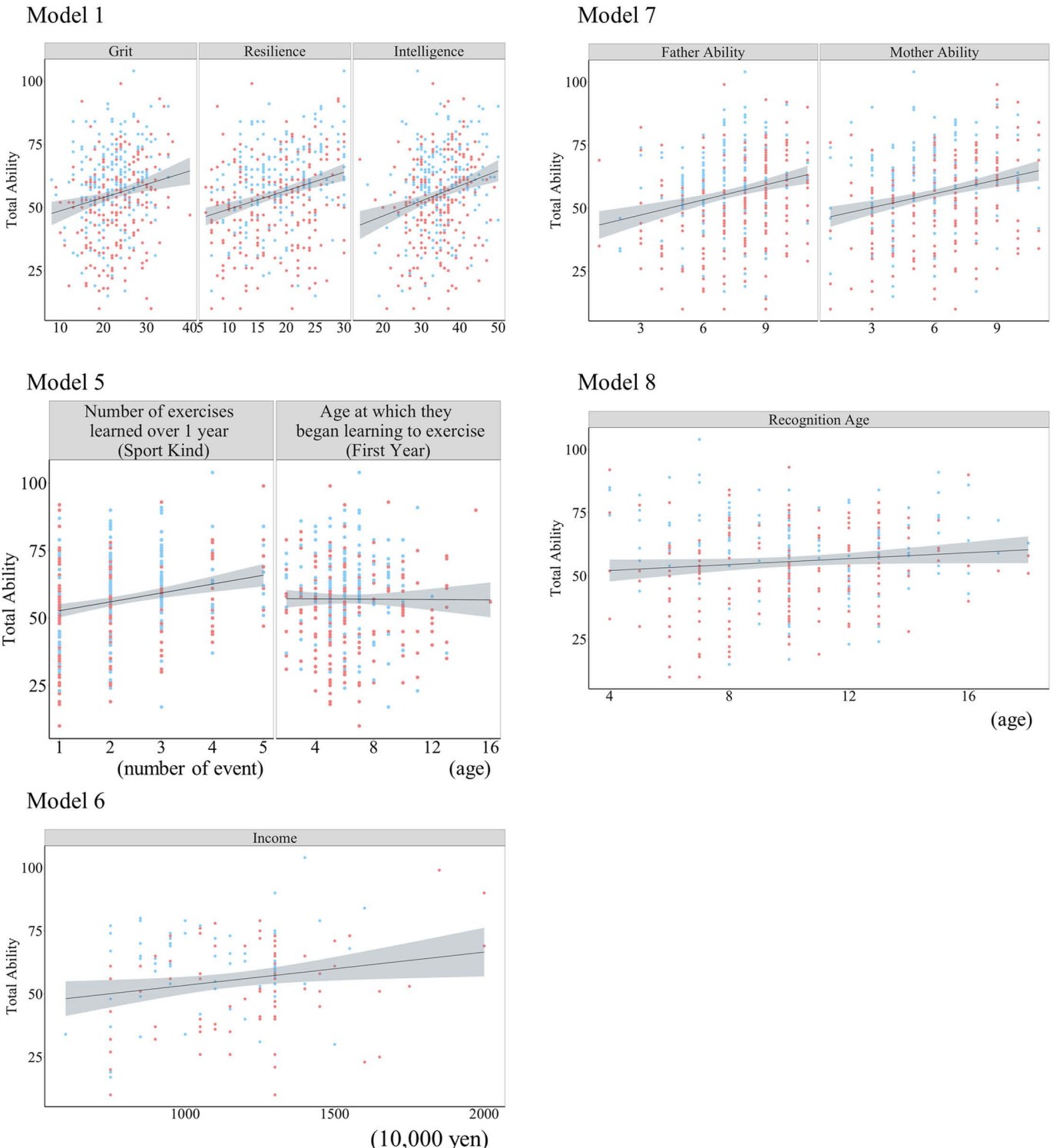

**Fig 2. Relationship between the explanatory variables used in each model and subjective total athletic ability.** The horizontal axis shows each explanatory variable, and the vertical axis shows subjective total athletic ability. The dots in the figure represent individual observations, with red and blue indicating females and males, respectively. The straight lines indicate regression lines based on the least-squares method, and the gray bands around the lines indicate 95% confidence intervals of the regression lines.

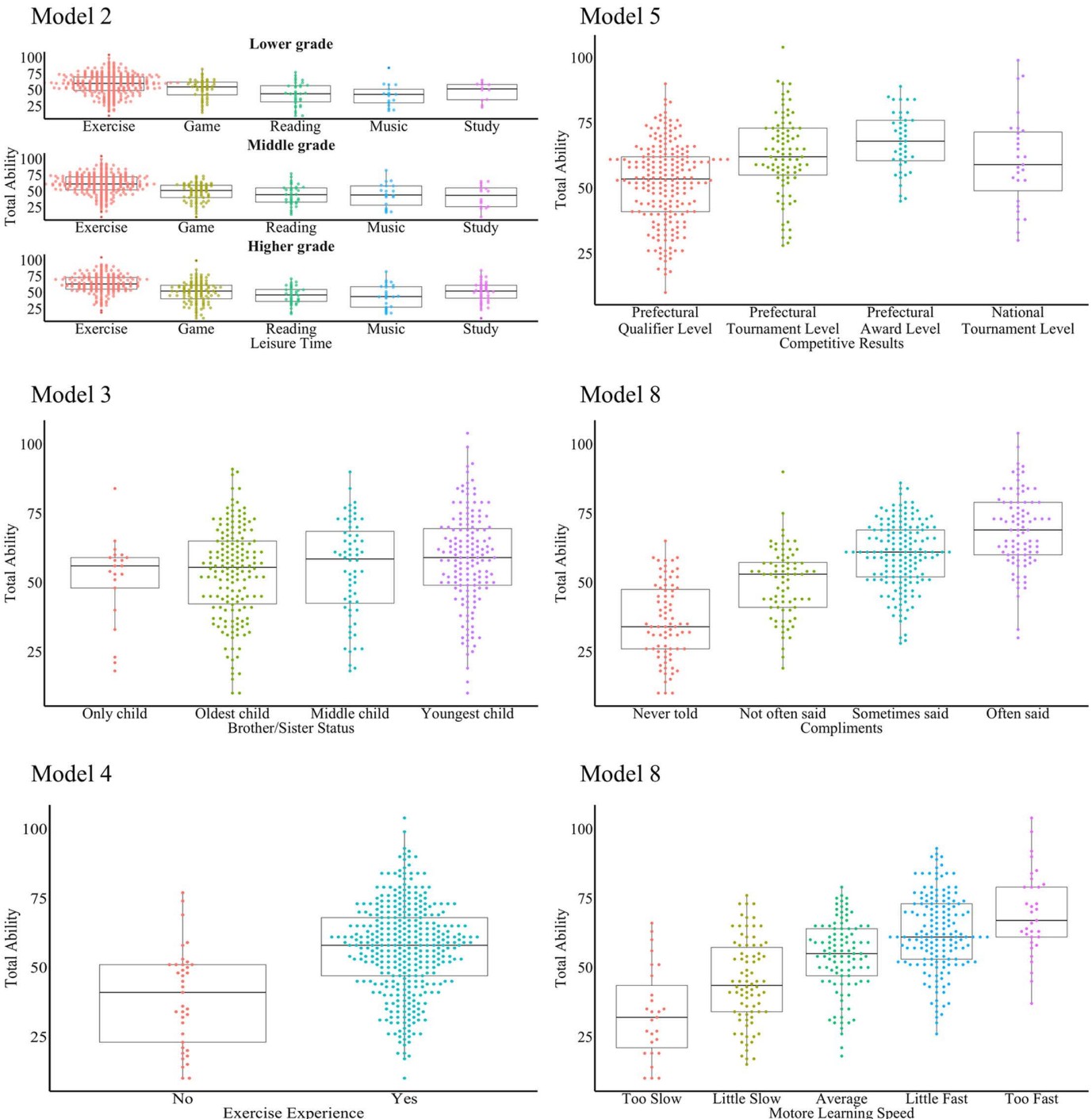

**Fig 3. Relationship between the categorical explanatory variables used in each model and subjective total athletic ability.**

**Table 3. Model assumptions and diagnostic test results.**

| Model | Durbin-Watson (DW) | Autocorrelation | Breusch-Pagan p-value | Homoscedasticity | Additional Notes |
|---|---|---|---|---|---|
| Model 1 | 1.9 | No autocorrelation | 0.41 | Confirmed | |
| Model 2 | 1.9 | No autocorrelation | 0.49 | Confirmed | |
| Model 3 | 1.97 | No autocorrelation | 0.83 | Confirmed | |
| Model 4 | 2.02 | No autocorrelation | 0.7 | Confirmed | |
| Model 5 | 2.08 | No autocorrelation | 0.18 | Confirmed | |
| Model 6 | 2.03 | No autocorrelation | 0.54 | Confirmed | |
| Model 7 | 1.82 | Marginally autocorrelation | 0.68 | Confirmed | Repeated Durbin-Watson test 10 times; all p > .05 |
| Model 8 | 2.02 | No autocorrelation | 0.25 | Confirmed | |

We found that an individual's exercise experience influenced their subjective total athletic ability. According to a study that investigated high and low motor competence, exercise habits, and health-related indicators among college students [32], students in the low motor competence group were more likely to have no experience with exercise club activities, whereas those in the high motor competence group were more likely to have exercise habits. Engaging in sports outside of physical education classes during regular school activities probably leads to improvements in physical fitness and athletic abilities, resulting in an improvement in total athletic ability. The various sports practiced for more than a year and the level of athletic performance was linked to total athletic ability rather than the age at which they first began participating in exercise or sports. A study [33] that investigated the effects of youth sports life careers on the frequency of sports in adulthood showed that adults who had played multiple sports during their elementary, junior high, and high school years played sports more frequently than those who had played a single sport. Therefore, the practice of multiple sports might have led to an increased frequency of sports in adulthood, and subjective total athletic ability improved as the participants experienced various physical activities and acquired sports skills.

Previous studies on short-distance racing have reported that children with better short-distance racing records have a higher sense of motor competence [34]. In short-distance racing, individuals can easily compare themselves with others in terms of their times and rankings. This could lead to a clearer evaluation of athletic ability. Consequently, short-distance racing is a sport in which athletic ability can be judged objectively based on the perceived difference in ability compared with others. A study of junior high school students [35] showed that despite improvements in short-distance racing records as they grew older, their rating scores for "good at short distance race," which indicates perceptions of physical competence, dropped significantly. Compared with other sports, it is relatively difficult to feel a genuine sense of improvement in skills, and easy to feel limited in improving records. This study suggests that individuals with high subjective abilities in short-distance races also exhibit high subjective abilities in other sports. However, compared with other sports, short-distance racing tends to reflects an individual's physical ability more than skill and experience. Further studies are required to determine the effects of short-distance racing ability on other sports.

A study [36] examined the overall ball gaming ability and motor skills of college students and reported that those who were good at ball games rated their motor skills as good. Subjective athletic abilities in soccer, volleyball, and basketball were highly correlated with subjective general athletic ability. Individuals with high athletic ability are likely to have a sense of athletic competence in popular sports, especially considering the intense competition within teams during club activities. Studies examining various sports and the relative age effect [20,22] have reported a strong effect in major sports but not minor sports. Relative age effects are strongly correlated with motor competence [37], and the sense of motor competence may be enhanced by being good at major sports, leading to improved ability in various sports (subjective general athletic ability).

The relationship between subjective general athletic ability and exercise category suggests that high motivation to improve motor skills is linked to high subjective athletic ability. Personality traits such as grit and resilience are thought to be the ability to sustain effort when placed in difficult situations in the process of achieving long-term goals [38]. High competence in these personality traits is considered the ability to persevere in acquiring superior skills. In addition, the personality trait of persistence is involved in maintaining a high level of motivation for improvement, which is necessary to consistently achieve goals. Furthermore, the view of intelligence examined in this study reflects a mindset that believes that talent can be changed through hard work [31]. We speculate that the influence of views of intelligence on subjective total athletic ability observed in this study may have resulted in higher subjective total athletic ability as a reflection of the ability to continue working hard to acquire advanced skills.

This study revealed the importance of leisure time in elementary school on subjective total athletic ability. Compared with physical activity, games, and studying in the middle grades, leisure activities other than exercise (games, reading, and music) in the higher grades negatively affected subjective total athletic ability. A study of lifestyle habits in various countries [39] reported that spending > 2 h daily playing video games decreased the habit of moderate to vigorous physical activity. Sedentary lifestyle habits such as television, video, and video games in elementary school children, body mass index, and abdominal circumference were significantly reduced in the intervention group that reduced these habits [40]. Therefore, leisure time spent sedentarily, such as playing games or reading books, leads to decreased subjective total athletic ability. Physical activity during childhood may be significant in the development of motor-skill proficiency in adulthood.

There was a positive association between being the youngest siblings and subjective total athletic ability. Previous studies with young children have reported that being with other children involves more physical activity than being alone among 3–6-year-olds [41] and that living with other children of the same age or older contributes to improved athletic ability [42]. Similarly, it has been shown that younger siblings (younger brothers/sisters) exhibit better athletic ability than only children [13]. Studies on preschool siblings have also reported that when older children learn to crawl, walk, and climb, younger children frequently imitate their actions [43]. These studies suggest that the youngest children who grew up in an environment in which siblings of the same or older age were always present at home were more physically active and had more opportunities to imitate their older siblings' movements, leading to a relative increase in their subjective total athletic ability.

Considering that sibling composition is associated with subjective total athletic ability, family involvement may be a significant factor in the development of motor skills. This study found that the subjective athletic abilities of the fathers and mothers were associated with their subjective total athletic abilities. The effect of parental athleticism on child athleticism occurs via both genetic and environmental processes. Studies examining the heritability of athletic performance have reported 56% heritability of maximal oxygen uptake [44] and 52% heritability of muscle strength and power [45]. This indicates that genetic factors explain over 50% of individual differences in athletic performance. Based on these studies, parents' athletic ability may have had a genetic influence on their children's athletic ability. Furthermore, Xu et al. [46] reported a positive correlation between parents' and children's physical activity levels and the influence of parents' perceptions of the importance of physical activity on children's physical activity. Parents with high athletic abilities may have a better understanding of the importance of physical activity for their children, potentially impacting their children's subjective total athletic ability.

The positive association between subjective total athletic ability and household income suggests that household economic status positively affected subjective total athletic ability. Previous studies have shown that higher socioeconomic status is associated with higher physical fitness levels [16] and that families with higher annual household incomes are more likely to engage their children in sports activities [47]. Therefore, the more economically affluent the family, the greater the opportunities for sports activities, the higher the level of physical fitness, and the greater the subjective total athletic ability.

Furthermore, the evaluation of good motor skills by others was associated with a higher subjective total athletic ability. The results obtained in this study indicate the importance of evaluations by others. Since positive feedback from others enhances intrinsic motivation [48], positive feedback on athletic ability probably triggers increased intrinsic motivation for exercise and improves subjective total athletic ability. However, it is important to acknowledge that there are various possible explanations for this association. It could be that both individuals and others are simply observing the same underlying ability objectively, with no direct influence between the two. Alternatively, others' views of one's ability may shape one's self-perception, as suggested by Cooley's looking-glass self theory of identity formation [49]. Since our study did not directly measure others' perceptions but instead relied on individuals' reports of what they perceive and recall about others' evaluations, it is also possible that participants' views of their own abilities influenced their recollections and perceptions of feedback they received, a phenomenon consistent with well-established psychological theories such as confirmation bias [50]. We also observed that the faster the speed of learning a movement (action/technique) for the first time, the higher the total athletic ability. Physical education and sports activities in schools and clubs often occur in groups. The experience of being able to learn new skills faster than others might contribute to the development of a positive attitude toward sports and improvement in subjective total athletic ability.

This study identified meaningful effect sizes in the regression analyses, with Cohen's $f^2$ values ranging from small to very large. Notably, Models 2 and 8 showed large or very large effect sizes, underscoring the substantial contribution of the factors examined to variations in subjective athletic ability. However, comparing these effect sizes with prior studies is challenging due to differences in methodologies, study populations, and measurement approaches. Many previous studies also lack detailed reporting of effect sizes. These limitations emphasize the need for a systematic meta-analysis that harmonizes methodologies and standardizes the reporting of effect sizes to facilitate meaningful comparisons. Future research should aim to address these gaps and provide a more unified understanding of the factors influencing athletic ability.

This study has some limitations. There may be some bias in the parents' income and sample because this study was conducted only with students at a particular college. Moreover, parents' athletic abilities affect their children's athletic abilities; however, this influence cannot be completely eliminated. In addition, this study was conducted with participants who subjectively evaluated their athletic abilities. The study's reliance on self-reported data and its exploratory design limit its ability to provide concrete, objective measures or developmental trajectories of athletic ability and motor skills. In the future, more detailed studies will be possible by combining objective evaluation indices. Furthermore, as this study used a cross-sectional design to obtain data, we could not interpret the causal relationship between the factors associated with subjective total athletic ability found in this study. A longitudinal study is needed to determine changes in total athletic ability to identify factors that may enhance motor skills. This study examined the factors associated with subjective total athletic ability by employing various types of linear models; however, it did not ascertain the extent to which each factor contributed to subjective total athletic ability. Therefore, additional studies are required to confirm these findings. Future research should aim to address these gaps by utilizing validated measures, conducting longitudinal studies, and incorporating objective performance metrics.

## 5. Conclusion

We aimed to identify the multiple factors associated with subjective total athletic ability (commonly referred to as motor skills) among undergraduate students. Our results indicated that various factors were associated with subjective total athletic ability, including personality traits such as grit, resilience, and a sense of intelligence; how they spend their leisure time; sibling composition; previous exercise experience; parents' athletic ability; household income; and how often they are called "athletic" by others. Therefore, it is necessary to clarify how these factors can determine good or bad motor skills from a comprehensive perspective. Based on the relationships between motor skills and the associated factors identified in this study, multiple factors may be intricately associated with the development of a sense of motor skills.

## Supporting information

**S1 File. All data.**
(XLSX)

**S2 File. R code.**
(DOCX)

**S3 File. Normality assumption and tables of Type III test results.**
(PPTX)

## Acknowledgments

We would like to thank all the participants of the study. We are grateful for this comment.

## Author contributions

**Conceptualization:** Sho Ito, Kouki Kato.

**Data curation:** Sho Ito, Keishi Soga, Kouki Kato.

**Formal analysis:** Sho Ito, Keishi Soga.

**Funding acquisition:** Kouki Kato.

**Investigation:** Sho Ito, Kouki Kato.

**Methodology:** Sho Ito, Keishi Soga, Kouki Kato.

**Project administration:** Sho Ito.

**Resources:** Sho Ito, Kouki Kato.

**Software:** Keishi Soga.

**Supervision:** Sho Ito, Keishi Soga, Kouki Kato.

**Validation:** Sho Ito, Keishi Soga.

**Visualization:** Keishi Soga.

**Writing – original draft:** Sho Ito, Keishi Soga, Kouki Kato.

**Writing – review & editing:** Sho Ito, Keishi Soga, Kouki Kato.

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
