## [Decision Letter · Decision Letter 0]

28 Nov 2024

PONE-D-24-20019Determinants of subjective total athletic abilityPLOS ONE

Dear Dr. Ito,

Thank you for submitting your manuscript to PLOS ONE. After careful consideration, we feel that it has merit but does not fully meet PLOS ONE’s publication criteria as it currently stands. Therefore, we invite you to submit a revised version of the manuscript that addresses the points raised during the review process.

I encourage you to address all the points raised by the reviewers. Reviewer 2’s embedded comments, provided in the attached PDF, offer additional guidance for refining the manuscript.

We look forward to receiving your revised manuscript.

Kind regards,

Sarah Liddle

Academic Editor

PLOS ONE

Journal requirements: When submitting your revision, we need you to address these additional requirements. 1. Please ensure that your manuscript meets PLOS ONE's style requirements, including those for file naming. The PLOS ONE style templates can be found at https://journals.plos.org/plosone/s/file?id=wjVg/PLOSOne_formatting_sample_main_body.pdf and https://journals.plos.org/plosone/s/file?id=ba62/PLOSOne_formatting_sample_title_authors_affiliations.pdf 2. Please include captions for your Supporting Information files at the end of your manuscript, and update any in-text citations to match accordingly. Please see our Supporting Information guidelines for more information: http://journals.plos.org/plosone/s/supporting-information.  3. We are unable to open your Supporting Information file [Motor_Skill_PlosOne_submit.R ]. Please kindly revise as necessary and re-upload. 4. Please review your reference list to ensure that it is complete and correct. If you have cited papers that have been retracted, please include the rationale for doing so in the manuscript text, or remove these references and replace them with relevant current references. Any changes to the reference list should be mentioned in the rebuttal letter that accompanies your revised manuscript. If you need to cite a retracted article, indicate the article’s retracted status in the References list and also include a citation and full reference for the retraction notice.

Reviewers' comments:

Reviewer's Responses to Questions

**Comments to the Author**

1. Is the manuscript technically sound, and do the data support the conclusions?

Reviewer #1: Yes

Reviewer #2: Partly

2. Has the statistical analysis been performed appropriately and rigorously? 

Reviewer #1: Yes

Reviewer #2: Yes

3. Have the authors made all data underlying the findings in their manuscript fully available?

Reviewer #1: Yes

Reviewer #2: Yes

4. Is the manuscript presented in an intelligible fashion and written in standard English?

Reviewer #1: Yes

Reviewer #2: Yes

5. Review Comments to the Author

Reviewer #1: BRIEF SUMMARY

This paper presents the findings about the determinants of subjective (perceived) athletic ability. The study aimed to conduct a questionnaire survey among college students to determine the relationship between their subjective general athletic ability (commonly referred to as motor skills) and their subjective athletic ability in various sports, personality traits, sibling composition, socio-economic status, and athletic performance.

The current manuscript does not replicate articles in English, and the few close articles on this topic can only be found in Japanese. The manuscript displays some degree of originality, making it potentially beneficial for the English-speaking community.

The weakness of the paper is that it is a descriptive cross-sectional study.

GENERAL COMMENTS FOR AUTHORS

ABSTRACT

In research, researchers often observe a group of participants over a long period of time, called a cohort. It is therefore inappropriate to use the term “cohort” in a cross -sectional study.

BACKGROUND OF THE STUDY

The literature in Japanese reflects research and approaches that go back more than ten years, but today's understanding may already be much more refined and profound. It would therefore be more useful to focus on this area, with an emphasis on recent publications.

A real rationale for the study is presented but no hypotheses are offered. As a result, it is not clear why the study should be conducted, how it is built on previous research (and overcomes some of the problems of previous work), what results are expected.

My principal recommendation is that the introduction should clearly state the research hypothesis(-es).

RESEARCH METHODS

The section is missing some important information.

Representativeness – statistical power (sample size calculation), should be presented.

Please describe inclusion and exclusion criteria.

Quantitative indices of instruments validity and reliability must be provided (not only Cronbach alphas but also Mc Donalds omegas).

Please indicate that study variables were checked for normality assumptions (for instance, using calculation of Skewness and Kurtosis for all study variables).

Have some assumptions for linear regression analysis been tested and met (For instance, independence (The Durbin-Watson Statistic), homoscedasticity (The Breusch-Pagan test))?

Why these and not others statistical methods (why the lm () function of R was used to perform linear regression analysis?), as well as a precise argumentation of possible alternatives justifying that they are less suitable than the present one.

RESEARCH RESULTS

Why different terms are used: subjective total athletic ability and subjective general athletic ability, subjective general athletic performance and subjective overall motor performance – as this can lead to confusion. It should be unified.

It is incorrect to use the term "total athletic ability" without the word "subjective".

DISCUSSION PART

Everything must be explained based on the answer to the research question (hypothesis), and in comparison, with other very significant and current studies (2023, 2022) should be clarified much more if the objective has been achieved and why.

Please discuss the effect sizes (R2 for regression analysis). Were the observed effects strong/mild/weak? Compare the effect sizes in your data with the effect sizes in previous studies if possible.

Reviewer #2: Although there are no fatal flaws, there are opportunities to clarify what the study actually achieves and adds to the corpus of scientific literature. The statistical methods could be improved to avoid merely performing numerous associational analyses with minimal structure and organization. The use of clearer language in terms of the constructs being investigated and the conclusions that can be supported by them (throughout the paper) would enhance its rigor. Please see the embedded comments on the manuscript for more details.

6. PLOS authors have the option to publish the peer review history of their article (what does this mean? ). If published, this will include your full peer review and any attached files.

**Do you want your identity to be public for this peer review?** For information about this choice, including consent withdrawal, please see our Privacy Policy .

Reviewer #1: No

Reviewer #2: No

---

## [Author Response · Author response to Decision Letter 1]

12 Dec 2024

Dear Reviewer,

We sincerely appreciate your valuable suggestions and thoughtful feedback on our manuscript. Your comments were insightful and greatly contributed to improving the clarity and rigor of our study.

We have carefully addressed each point you raised, dedicating significant effort to ensuring our responses were thorough and aligned with your recommendations. Where necessary, we also made additional modifications to enhance the overall quality of the manuscript.

Thank you again for your constructive input, which has been instrumental in refining our work. We are confident that the revised manuscript is much stronger as a result of your guidance.

Sincerely,

Sho ito

---

## [Decision Letter · Decision Letter 1]

20 Apr 2025

Determinants of subjective total athletic ability

PONE-D-24-20019R1

Dear Dr. Ito

We’re pleased to inform you that your manuscript has been judged scientifically suitable for publication and will be formally accepted for publication once it meets all outstanding technical requirements.

Kind regards,

Dr Nick Dobbin

Academic Editor

PLOS ONE

Additional Editor Comments (optional):

Dear Dr Ito,

Thank you for submitting your manuscript to PLOS ONE for consideration after revising the work based on the reviewer's feedback.

After careful evaluation, I am delighted to accept this manuscript. I believe there is some scope add a few more citations within the background and enhance the clarity of the figures, so please check during the proofing stage.

Best wishes,

Dr Nick Dobbin

Reviewers' comments:

None

Reviewer's Responses to Questions

Not Applicable 

**PLOS One Comments to the Author**

1. If the authors have adequately addressed your comments raised in a previous round of review and you feel that this manuscript is now acceptable for publication, you may indicate that here to bypass the “Comments to the Author” section, enter your conflict of interest statement in the “Confidential to Editor” section, and submit your "Accept" recommendation.

Reviewer #1: All comments have been addressed

2. Is the manuscript technically sound, and do the data support the conclusions?

Reviewer #1: Yes

3. Has the statistical analysis been performed appropriately and rigorously? 

Reviewer #1: Yes

4. Have the authors made all data underlying the findings in their manuscript fully available?

Reviewer #1: Yes

5. Is the manuscript presented in an intelligible fashion and written in standard English?

Reviewer #1: Yes

6. Review Comments to the Author

Reviewer #1: The authors did a good job and corrected everything according to the comments. The manuscript has been sufficiently improved. I recommend to publish the article.

7. PLOS authors have the option to publish the peer review history of their article (what does this mean? ). If published, this will include your full peer review and any attached files.

**Do you want your identity to be public for this peer review?** For information about this choice, including consent withdrawal, please see our Privacy Policy .

Reviewer #1: No

---

## [Editor Report · Acceptance letter]

PONE-D-24-20019R1

PLOS ONE

Dear Dr. Ito,

I'm pleased to inform you that your manuscript has been deemed suitable for publication in PLOS ONE. Congratulations! Your manuscript is now being handed over to our production team.

Kind regards,

on behalf of

Dr. Nick Dobbin

Academic Editor

PLOS ONE